# Delirium Stigma Among Healthcare Staff

**DOI:** 10.3390/geriatrics4010006

**Published:** 2018-12-31

**Authors:** Elizabeta B. Mukaetova-Ladinska, Glynis Cosker, Mahathir Chan, Michael Coppock, Ann Scully, Seon-Young Kim, Sung-Wan Kim, Richard J. Q. McNally, Andrew Teodorczuk

**Affiliations:** 1Institute of Neuroscience, Psychology and Behaviour, University of Leicester, Leicester LE1 7RH, UK; 2The Evington Centre, Leicestershire Partnership NHS Trust, Leicester General Hospital, Gwendolen Rd, Leicester LE5 4QG, UK; 3Northumberland, Tyne and Wear NHS Foundation Trust, Richardson Unit, Leazes Wing, Royal Victoria Infirmary, Newcastle upon Tyne NE1 4PL, UK; glynis.cosker@ntw.nhs.uk (G.C.); mahathir.chan@doctors.org.uk (M.C.); michael.coppock@ntw.nhs.uk (M.C.); ann.scully@ntw.nhs.uk (A.S.); 4Departments of Psychiatry, Chonnam National University Medical School, Gwangju 61469, Korea; sykimpsy@chonnam.ac.kr (S.-Y.K.); swkim@chonnam.ac.kr (S.-W.K.); 5Institute of Health & Society, Newcastle University, Sir James Spence Institute, Royal Victoria Infirmary, Newcastle upon Tyne NE1 4LP, UK; richard.mcnally@newcastle.ac.uk; 6School of Medicine and Health Institute for the Development of Education and Scholarship (Health IDEAS), Griffith University, Queensland 4122, Australia; a.teodorczuk@griffith.edu.au; 7The Prince Charles Hospital, Metro North, Brisbane, Queensland 4032, Australia

**Keywords:** delirium, stigma, medical professionals, nursing professionals, education, training

## Abstract

Older people with delirium occupy more than one third of acute medical beds and require increased medical attention, as care at present is suboptimal. In addition, since delirium is undetected, it should form a target for teaching in wards. Moreover, as people with delirium are largely dependent on daily interactions and care by inpatients professional staff, it is important to address stigmatisation of these vulnerable patients. This is especially important as previous studies have shown that negative staff attitudes towards these patients undermine good care. This single center cross-sectional study was designed to determine the extent of institutional stigma among health professionals involved in the care of people with delirium. For this, professional staff working on medical wards and in communities were approached to fill in a questionnaire containing the adapted Delirium Stigma Scale and the EuroQol five dimensions (EQ-5D-5L) questionnaire. Additional demographic information concerning their education and professional and personal experience with delirium was also collected. The characteristics associated with stigma were determined from the sample. The findings of our study provide an insight into the high level of stigmatisation of delirium patients among professionals (mean 11.66/18 points). This was not related to professionals’ own experiences of delirium, their educational and professional backgrounds, or them having received formal delirium education. However, working closely with people with delirium seems to have a positive impact on the de-stigmatisation of this population among health professionals. Our findings that attitudes are not influenced by formal delirium teaching need to be incorporated into the design of interprofessional educational interventions. Accordingly, we advocate more direct patient-oriented and care delivered teaching interventions.

## Key Points:

People with delirium occupy more than one third of acute medical beds and require increased medical attention and treatment.Professionals working in communities and medical wards have high levels of stigmatisation of delirium patients.Delirium stigmatisation was not related to professionals’ own experiences of delirium, their educational and professional backgrounds, or them having delirium education. However, working closely with people with delirium seems to have a positive impact on the de-stigmatisation of this population among health professionals.These findings need to be translated into a meaningful educational intervention with an emphasis placed on direct patient-oriented educational interventions.

## 1. Introduction

Delirium patients represent more than one third of all older people admitted to acute medical beds. The majority of them are inpatients in emergency and care of the elderly departments [1,2]. A number of publications have stressed the importance of the management of delirium in various medical settings [1,3,4]. Whereas research advancement has largely concentrated on causes, detection and management of delirium [5,6] and educational interventions [7], there is scant work on potential stigma experienced by people with delirium by professionals involved in their care. The latter is particularly important since people with delirium are largely dependent on daily interactions with medical, nursing, and other affiliated professionals who care for them during an episode of acute confusion. Our previous qualitative research has determined that negative attitudes towards patients with delirium by hospital staff undermine care processes [7]. 

Stigma is conceptualized as a complex social process co-occurring with labeling, stereotyping, undervaluation, separation, status loss, and discrimination [8,9]. Stigma arises out of normal human cognitive processes that evaluate threat and risk, organise social knowledge, and determine self-perception [10]. Since power is essential to the social production of stigma, with the label of ‘stigmatized person’ being likely given by those with power [8], it is not surprising that stigmatization occurs on multiple levels throughout the healthcare sector, such as structural, interpersonal, and intra-individual levels [9,11,12]. Although there are several studies that tackle the stigmatisation of people with mental health problems [13,14,15], there are very little data about the stigmatisation of mental health problems in older people by professionals, and they are confined to depression only [16,17,18] and assisted living [19]. Since stigma represents a major cause for health inequalities [20], identifying the factors that influence higher stigmatisation among health providers will result in better health provision [19]. Addressing attitudinal blocks to care will allow the attainment of higher level learning needs such as the detection of delirium [7].

In the case of older people with mental illnesses, stigmatisation results from ignorance and misconception of medical and mental health problems in old age, fear of injury, contamination, the burden of care, fear of one own’s ageing and maladies of old age, internalisation of stigmatising ideas (e.g., self-injury) by those affected (e.g., patients, their families, professionals), and complexity of the illness. A recent study conducted on people with dementia reported not only higher levels of burn-out among caregivers but they were also associated with more negative attitudes towards people with dementia [21]. In the lack of such studies in people with delirium, we cannot rule out that similar attitudes may have a serious impact on the delirium management, including non-recognition, misdiagnosis and inappropriate treatment, and a substantial negative impact on the patient quality of life, which is reviewed in [22,23]. At an organisational level, the stigmatisation of people with delirium may also lead to poor quality treatment and care (access, provision, outcome and relapse), marginalisation within care systems and pathways [24], demoralisation of professionals, problems with staff recruitment and retention, as well as negative impacts on delirium sufferers and their families. The latter includes unnecessary prolonged admissions and institutionalization, and poor quality of life.

In this study, we approached a variety of disciplines from different professional backgrounds that filled in a predesigned questionnaire containing demographic information, adaptation of the newly devised Delirium Stigma Scale (DSS) for use among professionals, and generic health status of delirium patients with the EuroQol five dimensions (EQ-5D) questionnaire comprising health state description and evaluation. 

## 2. Materials and Methods

### 2.1. Recruitment of Participants

Colleagues from 8 acute medical wards (dermatology, gastroenterology, cardiology, endocrinology, toxicology, trauma, burns unit, and renal medicine) and 5 care of elderly wards (including one care of the elderly rehabilitation ward), all working in the Newcastle Hospitals National Health Service (NHS) Foundation Trust, UK (Royal Victoria Infirmary, *n* = 49; Freeman Hospital, *n* = 41 (37 working in care of the elderly wards), a rehabilitation ward (*n* = 19)), were asked to voluntarily complete a questionnaire regarding their attitudes about people with delirium. Ward managers facilitated the distribution of the questionnaire among staff working on their wards. One week following the dissemination of the questionnaires, the hard copies of the questionnaires were collected. In addition, members of the Newcastle Integrated Liaison Service (*n* = 8), as well as participants to a regional delirium event (*n* = 40), were also approached to fill in the questionnaire. A total number of 200 questionnaires were distributed: 10 questionnaires per ward, including liaison service (*n* = 140 questionnaires) and 60 questionnaires distributed at the regional delirium event.

### 2.2. Assessment Scales

All participants were asked to fill in a 2 page questionnaire (Appendix A) containing the DSS, a modified rating tool based on the validated Perceived Stigma of Delirium Scale (PSDS) [25], and the physical health scale (EQ-5D), a standardized instrument for measuring generic health status, comprising of health state description and evaluation [the visual analogue scale to evaluate overall health status of delirium patients (EQ-VAS)] (http://euroqol.org/eq-5d-products/eq-5d-3l.html). EQ-5D was chosen as a “stated preference” (https://www.ohe.org/news/5-things-you-should-do-eq-5d-data) in order to collect views from participants regarding health problems in people with delirium. In addition, the questionnaire included general demographic details such as gender, working experience, educational level, personal experience of delirium, and information of formal delirium training over the last 5 years.

The reliability of the DSS scale for use in healthcare professionals was estimated similarly to the original paper for the PSDS scale using the Cronbach alpha. The DSS had similar reliability to the original PSDS scale [25]. Namely, the Cronbach alpha ranged from 0.570–0.728, and the internal consistency for the total score, Cronbach’s α, was 0.668 (somewhat lower than the original PSDS Cronbach alpha of 0.85). The DSS scale has only 6 items, and the lower Cronbach alpha results do not necessarily imply low reliability but do reflect the lower number of questions [26].

### 2.3. Statistical Analysis

Statistical analysis was conducted with SPSS v.24 (Armonk, NY: IBM Corp). Data were normally distributed and parametric tests were used for analysis. ANOVA was used to determine differences between distinct categories, and the Pearson correlation test was used to explore associations between continuous variables such as age, duration of delirium teaching (hours/5 years), and years of professional experience, as well as DSS, EQ-5D-5, and EQ-VAS scores. 

### 2.4. Ethical Approval

Since this work involves NHS staff recruited as research participants by virtue of their professional roles, it does not require Research Ethics Committees review within the UK Health Departments’ Research Ethics Service (http://www.hra.nhs.uk/documents/2013/09/does-my-project-require-rec-review.pdf).

## 3. Results

### 3.1. Participants’ Characteristics

A total of 157 participants (78.5% return rate) filled in a questionnaire containing information about delirium stigma and physical descriptions of patients’ with delirium, which included: nursing staff (12 ward sisters, 56 staff nurses, two advanced nurse practitioners, one community psychiatric nurse, one admiral dementia nurse, six student nurses), medical doctors (eight consultants, 13 trainee doctors), and other healthcare professionals (five social workers, five occupational therapists, two ward managers, three housekeepers, 21 health care assistants, three therapeutic support workers, four physiotherapists, two mental health lecturers, two podiatrists, two rehab technicians, one clinical and dementia educator, one delirium care dietician, one commissioner, one domestic assistant, one clinical lead, one pharmacy technician, and one award clerk). All ward-based participants were part of a clinical team, and as such were all included in the analysis of the data. The mean age of the participants was 40.03 years (standard deviation 11.83 years; range 18–60 years), with a mean professional experience of 11.87 years (standard deviation 10.01 years; range 0.08–37 years). Of them, 78 reported to have had a delirium experience themselves (49.7%), 64 (40.8%) had a close family member with delirium, and this in total represented 82 participants who declared either themselves and/or a family member to have had delirium (52.2%). Out of the 154 participants, 71 (45.22%) had attended formal delirium teaching. The mean delirium teaching over a five-year period was 1.71 h/years (standard deviation 3.2 h/year; range 0–20 h/year). 

### 3.2. Delirium Stigma

On average, participants had relatively high scores on the DSS scale (mean 11.66 out of maximum 18 points, ranging from 4–18 points), indicating high stigma. Only three (1.91%) participants had a DSS score ≤6/18 (a score indicative of a lack of stigma). Delirium stigma was not influenced by gender (Figure 1a), age and years of experience, professional role (Figure 1b), delirium experience (Figure 1c), or delirium education (Figure 1d). This was further supported when the participants were divided into three groups depending on their professional roles: medical staff, nursing staff, and non-medical, allied health professionals (F = 0.775, *p* = 0.463). However, professionals working in the liaison service, as well as those in geriatric medicine, showed significantly lower delirium stigma scores (means 10.63 and 10.87, respectively) compared to their counterparts working in general psychiatry or acute medicine (means 12.15 and 12.00; *p* = 0.036) (Figure 1e). These findings were due to the significantly lower scores obtained on five out of the six DSS questions (please note that only the DS4 item, “people with delirium are embarrassed or ashamed that they have experienced delirium” was similar among the analysed groups).

Similarly, a previous history of delirium (either participants’ own experiences or delirium experienced by a family member) did not influence the extent of delirium stigma (*p* = 0.437). In addition, the extent of formal delirium teaching did not influence the DSS. Thus, participants who had delirium teaching over the last five years (mean = 46.68 min/year) had similar DSS scores (mean = 11.62) to those who did not have formal delirium teaching (mean = 11.60; *p* = 0.959). Similarly, the professional background (medical/nursing vs. ward based vs. non-medical practitioners) had little influence on the DSS scores (*p* = 0.781) (Figure 1a,d).

### 3.3. Physical Health (EQ-5D-5L Analysis)

Participants’ attitudes towards the physical health of people with delirium were assessed with the EQ-5D-5L scale. The physical health was not influenced by gender (*p* = 0.348), previous history of delirium (*p* = 0.950), or formal teaching in delirium (Figure 1a,d). However, when analysed across different professional groups, the general professionals had somewhat higher scores on this scale in comparison with the other professionals. The groups did not differ among each other (*p* = 0.335) (Figure 1e). 

### 3.4. Physical Health with Delirium (EQ-VAS Analysis)

Neither gender nor professional contact with patients with delirium influenced the physical health with delirium scores (*p* = 0.799 and *p* = 0.565, respectively). In contrast, the participants commented that a previous episode of delirium, a professional background, and having formal delirium teaching influenced the severity of the physical health. Thus, general professionals felt that the physical health of people with delirium was significantly worse compared to other professionals, especially those working in geriatric medicine and acute medicine (*p* = 0.0001). Similarly, participants who had a personal experience of delirium or experience of delirium in their family reported significantly worse physical health in comparison to the group with no personal delirium experiences (*p* = 0.019). Formal delirium teaching influenced the delirium health observation, with subjects who attended the teaching sessions giving lower/more severe scores for delirium patients (mean = 30.38 vs. mean = 40.65, *p* = 0.003) (Figure 1a–e).

### 3.5. Correlation Analysis

The correlation analysis highlighted the close relationship between the DSS and physical health (*r* = 0.237, *p* = 0.009; Figure 2a), whereas the length of professional experience was associated with lower physical scores (*r* = −0.206, *p* = 0.024; Figure 2b). Not surprisingly, the length of professional experience was associated with participants’ ages (*r* = 0.677, *p* = 0.0001; data not shown). In contrast, the extent of delirium teaching did not influence either the DSS (*r* = 0.073, *p* = 0.395) or the physical health (*r* = 0.057, *p* = 0.539) scores, and was not linked to the length of professional experience (*r* = 0.043, *p* = 0.691) or age of participants (*r* = 0.014, *p* = 0.874; data not shown). However, it was associated with the objective measure of the physical health in delirium subjects, as measured with the EQ-VAS scale (*r* = −0.255; *p* = 0.004; Figure 1c). Thus, participants who had formal delirium training were able to correctly identify that patients with delirium had poor physical health.

## 4. Discussion

Our study revealed a surprisingly high degree of delirium stigmatization among all professionals, regardless of whether they worked directly with people with delirium or not. The extent of stigma was not related to participants’ ages, genders, education, or their personal experiences of delirium. However, professionals working in geriatric medicine and liaison psychiatry had significantly lower delirium stigma scores, although they still showed higher delirium stigma scores than the normal DSS scores. The score of the delirium scale (mean 11.7, interquartile range 10–13) by professionals in the present study was higher than the score of patients who experienced delirium (median 8.0, interquartile range 5–10) in a previous study using the same delirium scale [25]. This result suggests that professionals may not be able to help patients with delirium unless they listen to their experiences of stigma. Furthermore, specific professional development on stigma for professionals needs to be provided to reduce the risk of possible “iatrogenic stigma” [27]. The high delirium stigma scores appear to be driven by the professional perception of the physical state in subjects with delirium. Thus, the DSS and the physical scores were highly correlated, suggesting that the DSS may reflect the overall interviewer-rated degree of problem severity, similar to a self-perceived stigma and discrimination in people with coexistent mental and physical disability [28]. 

Similar findings of stigmatising attitudes were found in a recent systematic review on mainstream mental health professionals working with people with intellectual disability [29]. The study utilised a different methodological strategy that included a social-psychological triad of cognitive, affective, and behavioural dimensions, and reported a lack of familiarity and knowledge about people with intellectual disability. This was not the case for our participants’ knowledge about delirium. This argues that there may be additional factors (e.g., the misperception of older people’s health) that need to be explored in addition to those we examined in the current study.

There are only few studies addressing the health problems of health-care workers. The limited number of studies suggest that healthcare professionals are at a substantially higher risk of developing hypertension and cardiovascular diseases (three-fold higher than the general population [30]), stress [31], anxiety and depression [32,33], injuries [34], infectious disease [35], etc. In general, healthcare workers have high knowledge and positive attitudes but low compliance concerning standard precautions about health-care associated infections [36], thus they may well be at a higher risk of developing infections and other diseases that can easily result in acute confusion/delirium. In addition, we cannot exclude the possibility that the participants in our study may have been motivated to take part in the study by their own and/or family experiences with delirium, thus contributing to the higher percentage of delirium prevalence we describe. 

Our findings confirm that education is neglected among professionals working with older people with delirium. Thus, despite delirium being the most common hospital complication, up to 55% of participants did not have any formal delirium teaching within the last five years. Importantly, in the correlation analysis, educational background, in-depth education about delirium [37], and previous self-experience with delirium did not lower delirium stigmatisation. In fact, it was the length of professional experience and working with older people that had an influence on lowering delirium stigmatisation among professionals. These findings are surprising in light of the importance education plays in delirium prevention, treatment, and management [7]. This may be explained by the likely focus of educational approaches being on knowledge and skills rather than attitudes. This is an important finding given the results of the grounded theory study of learning needs, which identified that the teaching of delirium knowledge alone is likely to be ineffective unless attitudes are addressed first [7]. It also suggests that most formal teaching approaches are unlikely to factor in attitudes and stigmatisation as education targets. 

A recent study also highlighted the educational impact on improving professionals’ attitudes towards older people with cognitive impairment [38]. In that study, the educational intervention was of a prolonged nature repeated over three months with an overall duration of 250 min (fivefold longer than the formal delirium education the participants in this study had). This argues that longer educational interventions sustained over a period of time may have a greater impact on changing professional attitudes. 

In our study, delirium education correlated significantly with the knowledge about the physical health of people with delirium. This suggests that the two types of education—lifelong education (based on practical work with patients with delirium) and passive delirium education in a classroom—influence our professional attitudes toward delirium patients differently. Thus, longstanding work with people with delirium results in decreasing delirium stigma, whereas delirium educational trainings seem to increase the knowledge, but not the participants’ attitudes, towards these patients. In contrast, it seems that the professional experience based on clinical observational skills and cumulative professional knowledge working with older people have a significant impact on lowering the delirium stigma among medical professionals. This is further supported by our finding of the DSS score being lowest among professionals who work extensively with people with delirium, i.e., geriatric medicine and liaison psychiatry. In fact, previous studies showed that direct contact with the mentally ill was better than education at reducing stigma [39,40]. These data now need to be translated into meaningful educational interventions with an accent placed on direct patient-oriented educational interventions rather than brief teaching courses. The latter seem to be more beneficial in informing professionals about the delirium clinical presentation but do not influence their empathy towards delirium patients they work with. Furthermore, we advocate an interprofessional focus to such interventions and, where possible, involvement of caregivers and/or patients with an emphasis on the distressing nature of delirium to tackle the pertinent problem of attitudes. 

Our study has several limitations. In evaluating professionals’ attitudes towards delirium patients, we used a modified rating tool for use by professionals that captures the beliefs of the healthcare professionals. Stigma is detected from resulting behaviours. Such rating scales may not necessarily have the high sensitivity and specificity they have when used by patients themselves [22]. Furthermore, the DSS scores appear to be closely related to the medical knowledge of interviewed care professionals rather than delirium stigma, per se. In addition, we did not explore the affective dimension (e.g., fear, anxiety, lack of confidence), which could have also played a role in the high level of stigmatisation among medical professionals, as highlighted recently [29]. Another limitation of the study is the potential bias in participants’ recollections of the amount of teaching in delirium over the previous five years. This argues that novel tools need to be either developed or modified to address these caveats. However, a wide range of professionals were included in the study, confirming that the attitudes they have toward delirium patients are shared irrespective of the educational background, professional role, or direct care contact. In summary, our study casts an important light on healthcare professionals’ attitudes and stigma towards patients with delirium. We would like to stress that our findings do not provide evidence that high stigma, as rated on the DSS, is an indication that professionals would not provide high quality care to delirium patients. Our results suggest that attitudes are not influenced by formal delirium teaching. Arguably, teaching interventions should specifically focus on addressing stigma, potentially by means of involving patients in teaching processes and providing medical professionals with greater exposure to working with delirium patients. As attitudes ultimately block good delirium care, our findings are of importance to the successful management of patients with delirium. Further research should evaluate such attitudinal focused interventions.

## Figures and Tables

**Figure 1 geriatrics-04-00006-f001:**
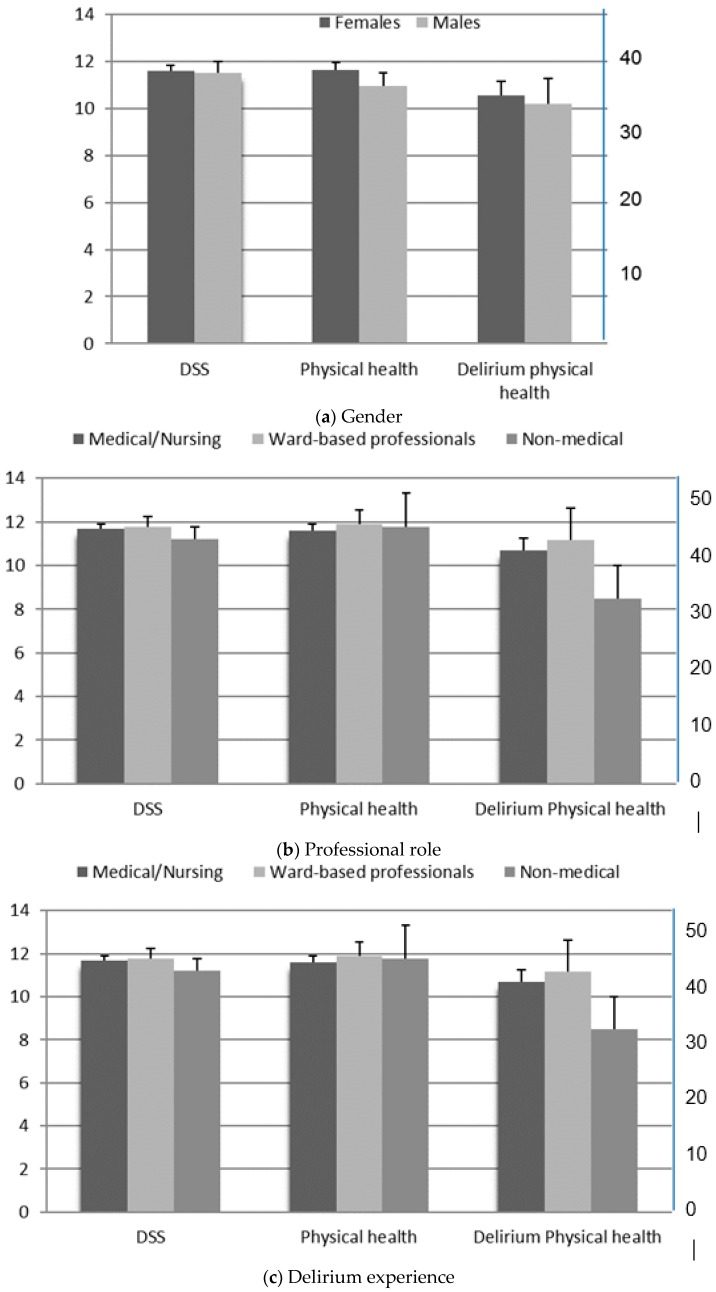
Analysis of delirium stigma, physical health, and delirium physical health measurements in respect to delirium in respect to (**a**) gender; (**b**) professional role; (**c**) delirium experience; (**d**) education; and (**e**) medical speciality. Physical health was determined according to EuroQol five dimensions questionnaire (EQ-5D-5L), and the delirium physical health was scored on the EQ-VAS. The scale on the left relates to the delirium stigma scale (DSS) and EQ-5D-5L measures (scores), whereas the one on the right is related to the EQ-VAS scores. Professionals working in geriatric medicine and liaison psychiatry had significantly lower DSS scores compared to professionals working in acute medicine and general psychiatry (**e**). * *p* < 0.05.

**Figure 2 geriatrics-04-00006-f002:**
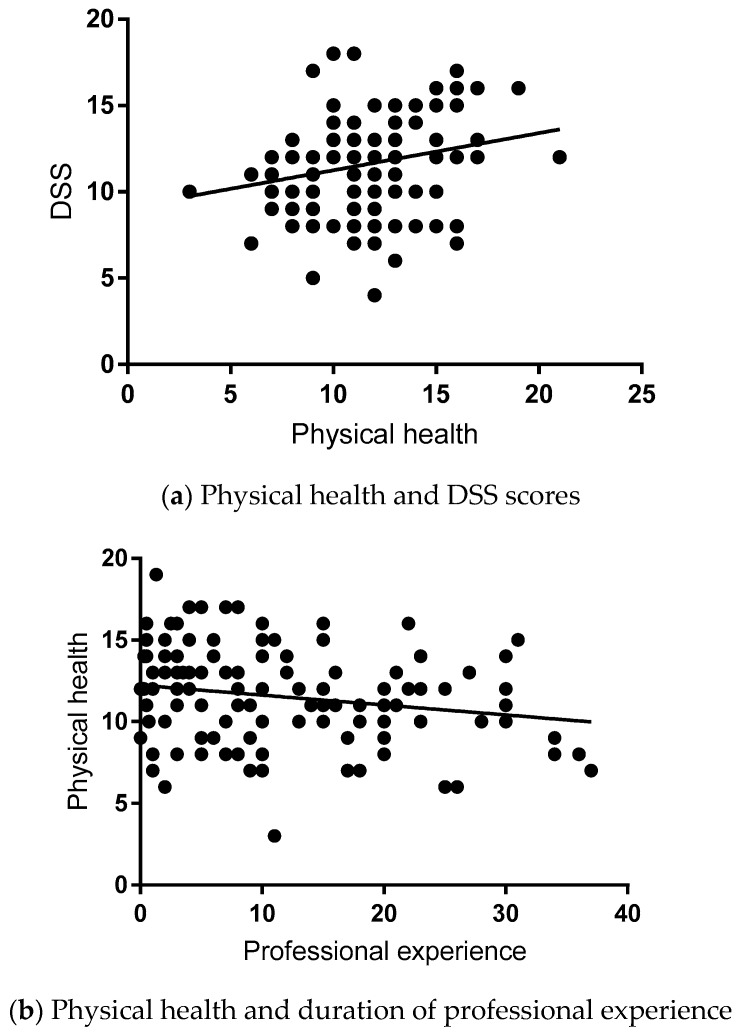
Relationship between physical health and the participants’ professional experiences and delirium educations. Physical health was assessed using the DSS, EQ-5D-5L, and EQ-VAS. Participants’ professional experiences were presented in years, delirium teaching refers to hours/five years period. Pearson correlation analysis was used. Statistical values (*r*- and *p*-values) were inserted within the main body of the text.

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
