# Peer review of "Delirium Stigma Among Healthcare Staff"

_geriatrics, 2018, doi:10.3390/geriatrics4010006_

Round 1

Reviewer 1 Report

Feedback for authors:

This project addresses an important issue that is extremely common in hospital settings.  The title is engaging and explanatory at the same time.  There are specific areas to consider for revision:

Abstract – please include Methods content, such as the study design, sampling process, primary outcomes, descriptive analysis, etc.  I was surprised that the conclusion in the abstract focuses on the need for education, as that didn’t seem to be mentioned at all in the abstract.  Perhaps if the methods described the analysis that characteristics associated with stigma were determined from the sample, that might be helpful.  It seems to be a suggestion for acting, rather than a conclusion from the study based on what you collected.

Introduction – it may help to explain how stigma differs from beliefs and attitudes.  Does a health professional having a negative attitude automatically lead to stigma?  Is it possible for a patient to be stigmatized if the healthcare professional has positive beliefs?  How are these related?  I ask because the tool later in the document seems to capture beliefs of the healthcare professionals, and stigma seem to be detected from resulting behaviours.  I would also suggest that paragraph 3 be further referenced as there are many general statements but it’s unclear if this is coming from the opinions of the authors or peer reviewed literature.

Methods – the sample (n) should be listed under Results.  It would also help to explain the tool and I could not open it in the appendix.  There is some detail about the tool in the text, which seems to support it as a valid tool, but it may help to explain why this versus any other tool had been chosen for this research.  Please also address the sampling approach – did anybody who came onto the unit meet criteria?  The focus of the title is nursing and medical staff, so I had expected to read about doctors and nurses, but there were a lot of different health professionals included.  It did not seem like purposeful sampling, but convenience, even including people that might not interact very often with patients directly, or deal with delirium firsthand.  Later in the results you describe ‘general’ health professionals, but it’s not clear which ones you categorized as general.  Also, explain how the people were invited – by poster or personal invites, by a research assistant on the unit?  Did they fill things out by hard copy or online?

Results – I was completely shocked to read that almost half of the sample had experienced delirium.  Given that it mostly occurs in frail older adults who are acutely ill, in ICU, or undergoing surgery, I wonder if this sample didn’t know what delirium actually was, or if the question could be interpreted in a strange way, and they considered ‘personal’ relating to something their family experienced.  Please explain in the analysis how you had the power to separate each healthcare profession.  It seems there were very few, and that type of subgroup analysis was not prespecified or accounted for in the sample size.   Also explain if the tool is validated to have each question analyzed separately, versus taken as a whole.  It appears that this led to a lot of analyses which may not have been powered.  For the figures it would be better to label the x and y axes on the chart instead of in the footnote.

Discussion – overall quite long but does address some important issues in this area.  The second last paragraph would logically be a bit earlier in the discussion.  I wonder if the issue of stigma relates more to ageism, and this literature could be brought into the debate about stigma related to delirium, or is it stigma related to older, frail seniors?  Also justify if you believe that every healthcare professional is equally important when it comes to delirium, as you analyzed them separately, but make general statements about the sample overall.

Author Response

This project addresses an important issue that is extremely common in hospital settings. The title is engaging and explanatory at the same time. There are specific areas to consider for revision:
Abstract – please include Methods content, such as the study design, sampling process, primary outcomes, descriptive analysis, etc.

Reply: We have added this information within the Method section of the abstract.

I was surprised that the conclusion in the abstract focuses on the need for education, as that didn’t seem to be mentioned at all in the abstract. Perhaps if the methods described the analysis that characteristics associated with stigma were determined from the sample, that might be helpful. It seems to be a suggestion for acting, rather than a
conclusion from the study based on what you collected.

Reply: This is a good point, and we have now included this in the abstract.

Introduction – it may help to explain how stigma differs from beliefs and attitudes. Does a health professional having a negative attitude automatically lead to stigma? Is it possible for a patient to be stigmatized if the healthcare professional has positive beliefs? How are these related? I ask because the tool later in the document seems to
capture beliefs of the healthcare professionals, and stigma seem to be detected from resulting behaviours.

Reply: This is a very interesting comment, and sadly we feel that addressing it fully is beyond the scope of the paper and our expertise. The reviewer is correct that the tool we used assess healthcare beliefs and these were used to determine the extent of stigma they exhibited. Similar tools have been used in other studies, and they all capture the beliefs of healthcare professionals, i.e. Gambling Perceived Stigma Scale (Donaldson et al, 2015), depression stigma scale (Vankar et al, 2014) etc.  This is now added in the discussion section.

I would also suggest that paragraph 3 be further referenced as there are many general statements but it’s unclear if this is coming from the opinions of the authors or peer reviewed literature.

Reply: We have amended this para as per reviewer’s comments.

Methods – the sample (n) should be listed under Results. It would also help to explain the tool and I could not open it in the appendix. There is some detail about the tool in the text, which seems to support it as a valid tool, but it may help to explain why this versus any other tool had been chosen for this research.

Reply: Thank you for this comment we have now moved the sample description within the result section, as suggested. We have used the tool, since there are not other similar tools developed for assessing stigma in people with delirium. Although there are a number of other stigma tools designed to address mental health stigma by healthcare professionals, including the recently introduced Opening Minds Stigma Scale for Health Care Providers (OMS-HC) (Modgill et al, 2014, BMC Psychiatry 14:120) none of them incorporates physical illness. We felt that for our study they were not applicable, and we modified the only available delirium stigma scale to assess healthcare professional attitudes. In the discussion we also discussed the limitation of the tool we used and call for further development of more specific tools.

Please also address the sampling approach – did anybody who came onto the unit meet criteria? The focus of the title is nursing and medical staff, so I had expected to read about doctors and nurses, but there were a lot of different health professionals included. It did not seem like purposeful sampling, but convenience, even including people that might not interact very often with patients directly, or deal with delirium first-hand.

Reply: We have addressed this in our reply to reviewer 2, and added the description of assessing professionals in the methodology section.

Later in the results you describe ‘general’ health professionals, but it’s not clear which ones you categorized as general. Also, explain how the people were invited – by poster or personal invites, by a research assistant on the unit? Did they fill things out by hard copy or online?

Reply: We have now addressed this. In the methodology section we escribed how participants were recruited. They were invited at a ward meeting to voluntarily participate in the questionnaire by the ward sister. They all filled in a hard copy – this information has now been added in the text.

Results – I was completely shocked to read that almost half of the sample had experienced delirium. Given that it mostly occurs in frail older adults who are acutely ill, in ICU, or undergoing surgery, I wonder if this sample didn’t know what delirium actually was, or if the question could be interpreted in a strange way, and they considered
‘personal’ relating to something their family experienced.

Reply: We appreciate that indeed there is a high number of participants who had experienced delirium. Please note that delirium is not confined to older people alone, and younger people, i.e. children and younger adults also experience delirium. The incidence of delirium among younger adults vary from 3% to over 50%, depending on the underlying physical illness and/or surgery. Post-operative, so called emergence delirium, is rather frequently reported among younger adults. One of the reasons for such a high delirium prevalence among our participants can be that only those with a personal or family member experiences took part in the study, or alternatively, that heath care professionals, because of the nature of their work, are exposed to higher exposure to factors predisposing them to delirium, i.e. infection, sleep deprivation, exhaustion etc. We have added this now in the discussion section.

Please explain in the analysis how you had the power to separate each healthcare profession.

Reply: For the analysis we grouped the participants into 3 groups: medical doctors (n=21), nursing staff (n=78) and other healthcare professionals (n=56). We have now grouped the professionals in the result section as above.

It seems there were very few, and that type of subgroup analysis was not prespecified or accounted for in the sample size. Also explain if the tool is validated to have each question analyzed separately, versus taken as a whole. It appears that this led to a lot of analyses which may not have been powered. For the figures it would be better to label the x and y axes on the chart instead of in the footnote.

Reply: This was covered in the methodology section when discussing the reliability of the DSS scale.  Please note that x and y axes are labelled in the chart (figure 2).

Discussion – overall quite long but does address some important issues in this area. The second last paragraph would logically be a bit earlier in the discussion.

Reply: We have now moved this para earlier in the discussion, as suggested.

I wonder if the issue of stigma relates more to ageism, and this literature could be brought into the debate about stigma related to delirium, or is it stigma related to older, frail seniors? Also justify if you believe that every healthcare professional is equally important when it comes to delirium, as you analysed them separately, but make general statements about the sample overall.

Reply: Thank you, the ageism is an important point when discussing stigma, and we have now included this.  We also discussed in the text the importance of the healthcare professionals, irrespectively of their qualifications and role.

Reviewer 2 Report

This manuscript examines the attitudes and stigma of health care workers toward patients with delirium. This is a very important topic with very little published in the current literature on this area. Please see comments and suggestion below for improving the manuscript

There are several typos and technical errors. See line 44 "the latter is in..." I think in needs to be deleted. Line 217 "sores" should be "scores"

Throughout the paper, the sentences are overly complex, making it very difficult to understand the points being made. I often had to reread sentences several times to understand the meaning. For example, line 45, "dependent on the daily interactions and the care medical" requires clarification with regard to which daily interactions and how are they dependent. How are the daily interactions different than the medical care which is provided. Line 195-196 it is unclear what is meant by "powered physical health"

Regarding the introduction, there is not enough evidence provided to support the idea that a high level of stigma results in poor delirium. More explanation is required as to how stigma interferes with care and what aspects of care may be affected.

In the methods, there needs to be a more detailed description of the stigma scale as well as the physical health scale. It was unclear how and why the original perceived stigma scale was modified. It would be helpful to see the stigma related questions. The relevance of the physical health score was unclear.

The conclusion is good overall, but again has many overly complicated and run-on sentences. It is certainly a strength of the study to include so many different disciplines, but that also makes the results difficult to interpret. When saying that liaison and geriatric specialists had lower stigma, were these groups M.D.s, nurses or other professions? It wasn't clear if these designations were based on training/expertise or work area. In terms of understanding the role of education, it is difficult to compare the education that would be provided to an MD vs an RN vs a nurses aid. The focus on recent, formal training on delirium is likely misleading, since the type of training is likely very different for each group of professionals.

Author Response

This manuscript examines the attitudes and stigma of health care workers toward patients with delirium. This is a very important topic with very little published in the current literature on this area. Please see comments and suggestion below for improving the manuscript

There are several typos and technical errors. See line 44 "the latter is in..." I think in needs to be deleted. Line 217 "sores" should be "scores"

Reply: Thank you, this is now amended.

Throughout the paper, the sentences are overly complex, making it very difficult to understand the points being made. I often had to reread sentences several times to understand the meaning. For example, line 45, "dependent on the daily interactions and the care medical" requires clarification with regard to which daily interactions and how are they dependent. How are the daily interactions different than the medical care which is provided. Line 195-196 it is unclear what is meant by "powered physical health"

Reply: Thank you for this comment. We have now gone through the text, and simplified the more complex sentences, and attended to wording in some sentences. Hope the text now reads better.

Regarding the introduction, there is not enough evidence provided to support the idea that a high level of stigma results in poor delirium. More explanation is required as to how stigma interferes with care and what aspects of care may be affected.

Reply: The reviewer is correct – indeed there are no studies on delirium and stigma, except the one from our Korean collaborators. We have instead expanded the introduction in relation to medical care of older people and included references in support of this.

In the methods, there needs to be a more detailed description of the stigma scale as well as the physical health scale. It was unclear how and why the original perceived stigma scale was modified. It would be helpful to see the stigma related questions. The relevance of the physical health score was unclear.

Reply: The stigma scale is shown in full in the APPENDIX, and the explanation for its modification is also enclosed. In addition, we provided an explanation for the physical health scale within the text in the methodology section.

The conclusion is good overall, but again has many overly complicated and run-on sentences. It is certainly a strength of the study to include so many different disciplines, but that also makes the results difficult to interpret. When saying that liaison and geriatric specialists had lower stigma, were these groups M.D.s, nurses or other professions? It wasn't clear if these designations were based on training/expertise or work area. In terms of understanding the role of education, it is difficult to compare the education that would be provided to an MD vs an RN vs a nurses aid. The focus on recent, formal training on delirium is likely misleading, since the type of training is likely very different for each group of professionals.

Reply: We have clarified this, and enclosed more information about the analysis, when distinct health professional groups were analysed. Interestingly, the education seems not to play a major role in the delirium stigma, since all health care professionals, irrespectively whether they are from medical, nursing or allied health care professional background had similar DSS scores. This has now been included in the result section.

Reviewer 3 Report

Mukaetova-Ladinska et al.: Delirium Stigma among Nursing and Medical Staff

Authors investigated medical professionals’ attitudes towards patients with delirium. This is an important issue as stigmatization can undermine good care. The study is well designed, correctly conducted, results are moderately interpreted. High level of stigmatisation was detected among health care professionals. Only working closely with people with delirium had a positive impact on the de-stigmatization of this population among health professionals.

Detailed comments:

Materials and Methods, 2.3 subchapter: The questionnaire is indicated to be in Appendix, but I couldn’t find it.

The study sample (n=157) was divided into 26 subgroups according to profession. There was a number of professional subgroup with only one representative. This number is too small for the comparison of different groups. Further, including non-medical professionals makes the sample more heterogeneous and the results less generalizable. I have doubts weather non-medical professionals are participating in delirium teaching et al. I suggest narrowing the sample to medical professionals. This narrowing will also strengthen the following conclusion of the authors: ’teaching interventions should specifically focus on addressing stigma potentially by means of involving patients in teaching processes and greater exposure of medical professionals to working with delirium patients.’

Results: page 8: I suggest omitting chart c from Figure 2, as it shows the evidence that professional experience is increasing with increasing age.

Reference list: Ref. No 7 and 20 is the same. Please delete No 20 and modify the citation numbers in the text!

Author Response

Authors investigated medical professionals’ attitudes towards patients with delirium. This is an important issue as stigmatization can undermine good care. The study is well designed, correctly conducted, results are moderately interpreted. High level of stigmatisation was detected among health care professionals. Only working closely with people with delirium had a positive impact on the de-stigmatization of this population among health professionals.

Detailed comments:

Materials and Methods, 2.3 subchapter: The questionnaire is indicated to be in Appendix, but I couldn’t find it.

Reply: The questionnaire is included in the APPENDIX but was not labelled properly. This has been now amended, and we also included labelling for the assessment of the physical state we used.

The study sample (n=157) was divided into 26 subgroups according to profession. There was a number of professional subgroup with only one representative. This number is too small for the comparison of different groups. Further, including non-medical professionals makes the sample more heterogeneous and the results less generalizable. I have doubts weather non-medical professionals are participating in delirium teaching et al. I suggest narrowing the sample to medical professionals. This narrowing will also strengthen the following conclusion of the authors: ’teaching interventions should specifically focus on addressing stigma potentially by means of involving patients in teaching processes and greater exposure of medical professionals to working with delirium patients.

Reply: We thank the reviewer for this comment. In the UK and Australia, the healthcare providers must ensure all staff are qualified, experienced and competent. As those working in medical environment often go beyond the role of nursing to provide extra support and care to patients, it is especially crucial that they all understand aspects such as infection control, prioritising health and wellbeing of both staff and patients in the delivery of services, food and fire safety etc. To ensure the provision of a continuously improving, safe and person centred working environment, all employees should undertake statutory and mandatory training appropriate to their role on entry to the organisation (Induction) and at regular intervals during their career. Another argument to include non-medical staff in the analysis is that their attitude(s) towards delirium was similar to that of medical professionals (nursing and medical staff), suggesting that delirium is ‘everybody business’. This inclusiveness provides further support of the statement about wider teaching of delirium among healthcare providers, irrespectively of their professional background. Interestingly, when divided into 3 groups, medical and nursing staff and allied health professionals, the DSS score did not differ among the 3 groups. This has now been included in the result section.

Results: page 8: I suggest omitting chart c from Figure 2, as it shows the evidence that professional experience is increasing with increasing age.

Reference list: Ref. No 7 and 20 is the same. Please delete No 20 and modify the citation numbers in the text!

Reply: We have omitted the reference.

Reviewer 4 Report

Thank you for giving me the opportunity to review this research paper called “

Delirium Stigma among Nursing and Medical Staff”. the topic is fascinating.

Delirium and the related stigma is an increasing issue in both acute and chronic settings.

The paper is well written and easily readable. The convenience sample may have reduced the relevance of results, but it could be implemented in further research to confirm Authors conclusions.

Author Response

Thank you for giving me the opportunity to review this research paper called “

Delirium Stigma among Nursing and Medical Staff”. the topic is fascinating.

Delirium and the related stigma is an increasing issue in both acute and chronic settings.

The paper is well written and easily readable. The convenience sample may have reduced the relevance of results, but it could be implemented in further research to confirm Authors conclusions.

Reply: Thank you for your comments.

Round 2

Reviewer 2 Report

The abstract is much improved in terms of sentence structure and readability. The appendix information is very useful to include. However, this raises some additional concerns about the overall soundness of the design and measurements used. It is not clear how the questions included in the DSS would indicate stigma against delirium patients. Additional background information is required to explain what is meant by stigma and how these questions are meant to gauge stigma against delirium. Delirium is a serious medical condition that can be a disturbing and frightening experience for patients. Therefore a belief that patients would be anxious about having delirium or that delirium would reoccur seem reasonable and I don't understand how that represents bias against patients. Likewise patient who have delirium are often quite physically ill during the course of delirium. So believing that people with delirium have poor physical health would again seem reasonable, rather than an indication of stigma. As stated in the previous comments to the authors, there is not enough evidence provided that high stigma as rated on the DSS would be an indication that professionals would not provide high quality to care to delirium patients.

Author Response

We thank reviewer 2 for the above comments. We have now amended the text as follows:

We     have added a brief para in the introduction regarding mental health stigma     (definition) and importance of stigma for healthcare, supported by     relevant references in the field;

In     the APPENDIX section we included now a brief description regarding the DSS     scale, providing an explanation of the development of the Perceived Stigma     of Delirium Scale on which the scale was based. This also provides an     explanation why a question regarding the delirium features in the DSS ;

The     stigmatisation as a result of delirium was explored further via using the     knowledge of health professionals about delirium (as assessed with the     Physical health scale EQ-5D). Their positive correlation indicates one of several     limitations that the DSS has, and we have addressed this in the     discussion. We also indicated in the discussion that the high DSS scores     are not indicative of poor medical care people with delirium obtain whilst     inpatients. In the discussion, we also call for further work to develop     novel delirium specific stigma scales that will encapsulate both the     healthcare professionals’ attitudes, believes and knowledge.
